# The *yellow* gene influences *Drosophila* male mating success through sex comb melanization

Jonathan H Massey[1,2], Daayun Chung[1], Igor Siwanowicz[2], David L Stern[2]*, Patricia J Wittkopp[1,3]*

[1]Department of Ecology and Evolutionary Biology, University of Michigan, Ann Arbor, United States; [2]Janelia Research Campus, Howard Hughes Medical Institute, Ashburn, United States; [3]Department of Molecular, Cellular, and Developmental Biology, University of Michigan, Ann Arbor, United States

**Abstract** *Drosophila melanogaster* males perform a series of courtship behaviors that, when successful, result in copulation with a female. For over a century, mutations in the *yellow* gene, named for its effects on pigmentation, have been known to reduce male mating success. Prior work has suggested that *yellow* influences mating behavior through effects on wing extension, song, and/or courtship vigor. Here, we rule out these explanations, as well as effects on the nervous system more generally, and find instead that the effects of *yellow* on male mating success are mediated by its effects on pigmentation of male-specific leg structures called sex combs. Loss of *yellow* expression in these modified bristles reduces their melanization, which changes their structure and causes difficulty grasping females prior to copulation. These data illustrate why the mechanical properties of anatomy, not just neural circuitry, must be considered to fully understand the development and evolution of behavior.

DOI: https://doi.org/10.7554/eLife.49388.001

*For correspondence:
sternd@janelia.hhmi.org (DLS);
wittkopp@umich.edu (PJW)

**Reviewing editor:**
K VijayRaghavan, National Centre for Biological Sciences, Tata Institute of Fundamental Research, India

## Introduction

*"The form of any behavior depends to a degree on the form of the morphology performing it."* – *West-Eberhard (2003)*

Over 100 years ago in Thomas Hunt Morgan's fly room, Alfred Sturtevant described what is often regarded as the first example of a single gene mutation affecting behavior (*Sturtevant, 1915*; reviewed in *Drapeau et al., 2003*; *Cobb, 2007*; *Greenspan, 2008*): he noted that *yellow* mutant males, named for their loss of black pigment that gives their body a more yellow appearance (*Figure 1A*), mated successfully with wild-type females much less often than wild-type males. In 1956, in what is often considered the first ethological study (reviewed in *Cobb, 2007*; *Greenspan, 2008*), Margaret Bastock compared courtship of *yellow* mutant and wild-type males and concluded that despite all courtship actions being present, loss of *yellow* function likely reduces courtship vigor or drive, leading to copulation inhibition (*Bastock, 1956*). Despite more recent data consistent with this hypothesis (*Drapeau et al., 2003*), the precise mechanism by which the *yellow* gene affects male mating success in *D. melanogaster* has remained a mystery. Consequently, Bastock's statement about *yellow* from her 1956 paper is equally true today: *"It seemed worthwhile therefore to examine more closely one example of a gene mutation affecting behavior and to ask two questions, (1) how does it bring about its effect? [and], (2) what part might it play in evolution?"*

The *D. melanogaster yellow* gene encodes a protein hypothesized to act either structurally (*Geyer et al., 1986*) or enzymatically (*Wittkopp et al., 2002*) in the synthesis of dopamine melanin, and a Yellow homolog has been shown to bind dopamine and other biogenic amines in the sand fly

**eLife digest** More than 100 years ago, Nobel-prize winning geneticist Thomas Hunt Morgan and his colleagues discovered that some fruit flies inherited genetic mutations that caused their body color to change. The yellow flies had a mutation in one specific gene and these mutants did not only look different from normal flies, they behaved differently too. Specifically, yellow males were far less successful at mating than normal males, demonstrating for the first time that some behaviors had a genetic basis.

Since then it has remained a mystery how the genetic mutations that cause yellow coloration in fruit flies lead to unsuccessful mating attempts. Geneticists have long suggested that mutations in insect pigment genes cause changes in the fly's brain because these pigments are made from dopamine, a chemical messenger that acts in the brain. They proposed that yellow flies must have altered levels of dopamine in their brains which was causing them to fail at mating.

To solve this mystery, Massey et al. used a series of genetic experiments and high speed-videos to assess how mutations in male yellow fruit flies affected their mating behavior. The experiments showed that yellow fruit flies mated poorly not because of changes in their brain but because of changes in specialized structures on their legs called sex combs. The yellow males lack melanin pigments in their sex combs, which changes their structure. As a result, the yellow males would court female flies but were then unable to grab and mount them. This explains why yellow flies often fail to mate and why fruit flies have sex combs in the first place.

The study reveals the importance of scientists considering that genes that affect behavior may do so by changing anatomy rather than by altering the brain. The results also may benefit those working to control insect pests. For example, they could help insect pest managers to develop strategies that prevent reproduction in other insects that spread disease or destroy crops.

DOI: https://doi.org/10.7554/eLife.49388.002

*Lutzomyia longipalpis* (*Xu et al., 2011*). The interaction between Yellow and dopamine might explain the protein's effects on male mating success because dopamine acts as a modulator of male courtship drive in *D. melanogaster* (*Zhang et al., 2016*). These effects of dopamine are mediated by neurons expressing the gene *fruitless* (*fru*) (*Zhang et al., 2016*), which is a master regulator of sexually dimorphic behavior in *D. melanogaster* that can affect every component of courtship and copulation (reviewed in *Villella and Hall, 2008*). *fru* has also been shown to regulate expression of *yellow* in the central nervous system (CNS) of male *D. melanogaster* larvae (*Drapeau et al., 2003*). These observations suggest that the pleiotropic effects of *yellow* on male mating success might result from effects of *yellow* in the adult CNS, particularly in *fru*-expressing neurons. Consistent with this hypothesis, functional links between the pigment synthesis pathway and behavior mediated by the nervous system have previously been reported for other pigmentation genes (*Hotta and Benzer, 1969*; *Heisenberg, 1971*; *Borycz et al., 2002*; *Richardt et al., 2002*; *True et al., 2005*; *Suh and Jackson, 2007*).

## Results and discussion

### Fruitless-expressing cells do not mediate the effect of yellow on male mating success

*D. melanogaster* males perform multiple behaviors, including tapping, chasing, singing, and genital licking, before attempting to copulate with females by curling their abdomen and grasping the female (*Figure 1B*, *Video 1*). In one-hour trials, we found that virgin males homozygous for a null allele of the *yellow* gene ($y^1$) successfully mated with wild-type virgin females only 3% of the time, whereas wild-type males mated with wild-type virgin females 93% of the time (Fisher's exact test, p=$6\times10^{-13}$; *Figure 1C*). Videos of mating trials (e.g., *Videos 1* and *2*) indicated that the difference in mating success between wild-type and *yellow* males did not come from differences in the amount of time spent courting (courtship index, t-test, p=0.81; *Figure 1D*) or the number of wing extensions during the trial period (t-test, p=0.37; *Figure 1E*). Courtship song analysis also indicated similar amounts of pulse (t-test, p=0.90; *Figure 1F*), sine song (t-test, p=0.07; *Figure 1G*), and interpulse

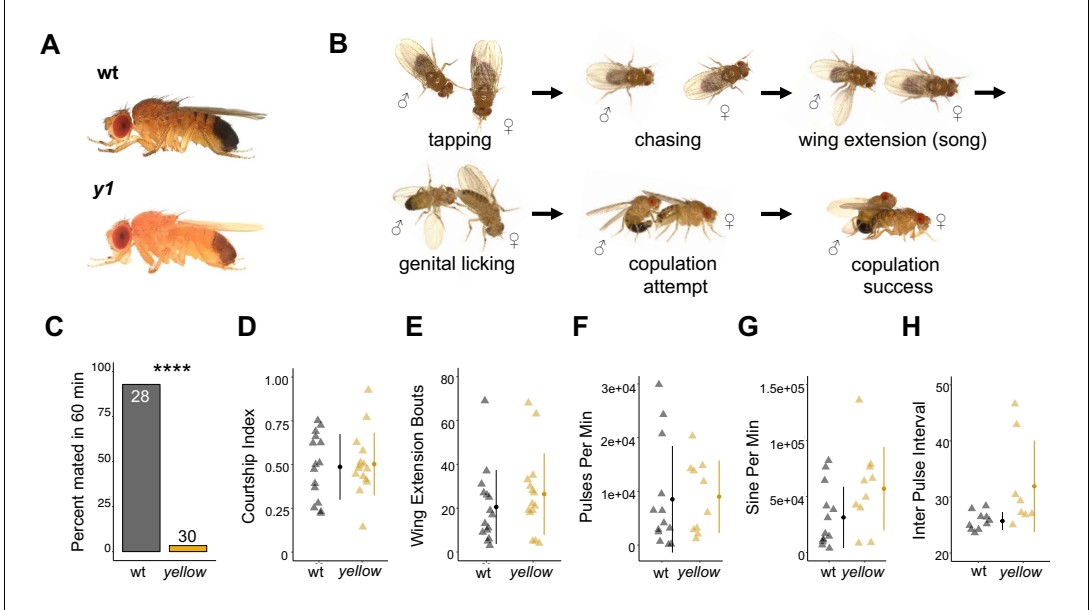

**Figure 1.** The *Drosophila melanogaster yellow* gene is required for male mating success. (A) Photographs comparing wild-type and *yellow* (y[1]) body pigmentation [Reprinted from Atlas of Drosophila Morphology, 1 st Edition, Sylwester Chyb and Nicolas Gompel, Body Markers, pp.173, 175, Copyright (2013), with permission from Elsevier] This panel is not covered under the CC-BY 4.0 licence. (B) Snapshots from videos illustrating *D. melanogaster* courtship behaviors. (C) y[1] males (yellow) showed significantly lower mating success levels compared to wild-type males (black) in non-competitive, one-hour trials. Sample sizes are shown at the top of each barplot. (D–H) y[1] males showed similar levels of courtship activity and song compared to wild-type males. (D) Courtship index: the proportion of time a male engages in courtship activity divided by the total observation period. (E) Wing extension bouts: the number of unilateral wing extensions during the observation period. (F) Pulses per minute. (G) Sine per minute. (H) Inter pulse interval. (D–H) Show individual points that represent single fly replicates. Circles represent means and lines SD. Significance was measured using Fisher's exact test in (C) and Welch's Two Sample t-test in (D–H). Comparisons that were statistically (p<0.05) are indicated (****p<0.0001).
DOI: https://doi.org/10.7554/eLife.49388.003

interval (t-test, p=0.07; *Figure 1H*). Watching the courtship videos showed that copulation initiation was most strikingly different between the two genotypes, with copulation initiation reduced in *yellow* males compared to wild-type (compare *Videos 3* and *4*).

To determine whether *yellow* activity in *fru*-expressing cells was responsible for this difference in mating success, we used the UAS-GAL4 system (*Brand and Perrimon, 1993*) to drive expression of *yellow-RNAi* (*Dietzl et al., 2007*) with *fru*[GAL4] (*Stockinger et al., 2005*), knocking down native *yellow* expression in these cells. We also used *fru*[GAL4] to drive *yellow* expression in y[1] mutants. In both cases, when the experimental genotype was compared to the (1) GAL4 only and (2) UAS only control

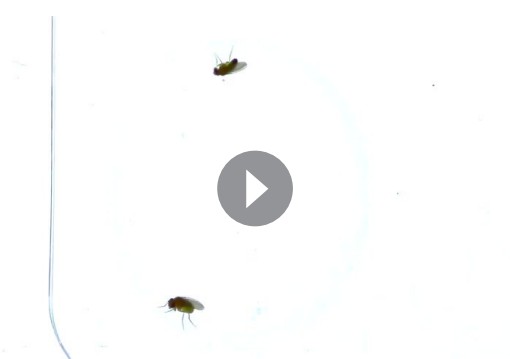

**Video 1.** Wild-type courtship and copulation.
DOI: https://doi.org/10.7554/eLife.49388.004

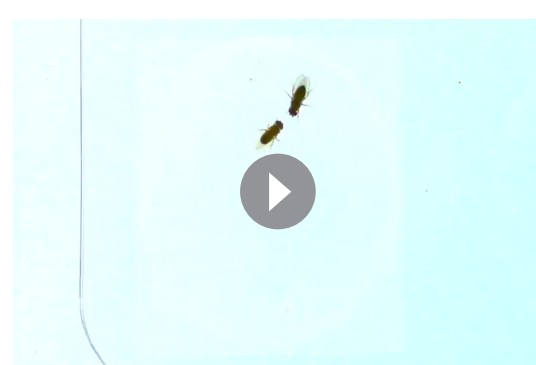

**Video 2.** y[1] courtship with wild-type female.
DOI: https://doi.org/10.7554/eLife.49388.005

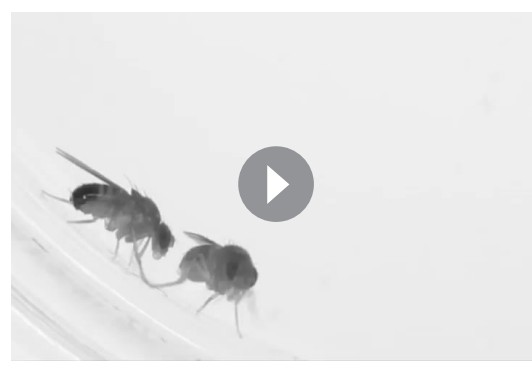

**Video 3.** Wild-type copulation.
DOI: https://doi.org/10.7554/eLife.49388.006

genotypes using a Fisher's exact test (FET) with p-values adjusted (Bonferroni) for the (n = 2) control comparisons, we found no significant effect on male mating success (*Figure 2A*, p=1 for both tests; *Figure 2B*, p=0.07 and 0.2), suggesting that expression of *yellow* in *fru*-expressing cells is neither necessary nor sufficient for *yellow*'s effect on male mating success.

## Doublesex-expressing cells require yellow for normal male mating success

To continue searching for cells responsible for *yellow*'s effects on mating, we examined a 209 bp sequence 5' of the *yellow* gene called the 'mating-success regulatory sequence' (MRS) in a prior study that reported it was required for male mating success (*Drapeau et al., 2006*). We hypothesized that the MRS might contain an enhancer driving *yellow* expression and found that ChIP-seq data indicate the Doublesex (Dsx) transcription factor binds to this region in vivo (*Clough et al., 2014*). Like *fru*, *dsx* expression is required to specify sex-specific behaviors in *D. melanogaster* (*Rideout et al., 2010*; *Robinett et al., 2010*; reviewed in *Villella and Hall, 2008*; *Yamamoto and Koganezawa, 2013*), suggesting that *yellow* expression regulated by Dsx through the MRS enhancer might be responsible for its effects on male mating behavior. We found that reducing *yellow* expression in *dsx*-expressing cells with either of two different $dsx^{GAL4}$ drivers (*Robinett et al., 2010*; *Rideout et al., 2010*) strongly reduced male mating success (*Figure 2C*, FET, $p=7\times10^{-9}$ and $1 \times 10^{-7}$; *Figure 2—figure supplement 1A*, FET, p=0.002 and 0.002), whereas restoring *yellow* activity in cells expressing $dsx^{GAL4}$ in $y^1$ mutants significantly increased male mating success compared with $y^1$ controls (*Figure 2D*, FET, p=0.001 and 0.0004; *Figure 2—figure supplement 1B*, FET, $p=5\times10^{-10}$ and $5 \times 10^{-10}$). Video recordings of male flies with reduced *yellow* expression in *dsx*-expressing cells showed the same mating defect observed in $y^1$ mutants: males seem to perform all courtship actions normally, but repeatedly failed to copulate (*Video 5*). We therefore conclude that *yellow* expression is required in *dsx*-expressing cells for normal male mating behavior.

To determine whether the MRS sequence might be the enhancer mediating *yellow* expression in *dsx*-expressing cells that affects male mating success, we manipulated *yellow* expression with GAL4 driven by a 2.7 kb DNA region located 5' of *yellow* that includes the wing, body, and putative MRS enhancers (*Gilbert et al., 2006*, *Figure 2—figure supplement 2A*). Altering *yellow* expression with this GAL4 driver modified pigmentation as expected but did not affect male mating success (*Figure 2—figure supplement 2B–D*), possibly because this GAL4 line did not show any detectable expression in the adult CNS (*Figure 2—figure supplement 2E*). To test more directly whether the MRS was necessary for male mating success, we deleted 152 bp of the 209 bp MRS sequence using CRISPR/Cas9 gene editing (*Bassett et al., 2013*) (*Figure 2—figure supplement 2F,G*). We found that this deletion had no significant effect on male mating success (*Figure 2—figure supplement 2H*, FET, p=0.99 compared to wildtype (CS)), contradicting the previous deletion mapping data (*Drapeau et al., 2006*). We conclude therefore that effects of *yellow* expression in *dsx*-expressing cells on mating behavior are likely mediated by other *cis*-regulatory sequences associated with the *yellow* gene.

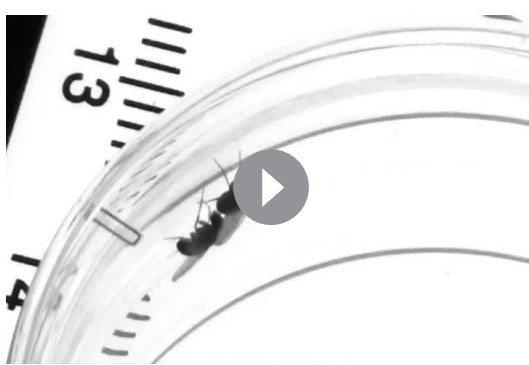

**Video 4.** Copulation attempts between $y^1$ male and wild-type female after 3 hr of courtship.
DOI: https://doi.org/10.7554/eLife.49388.007

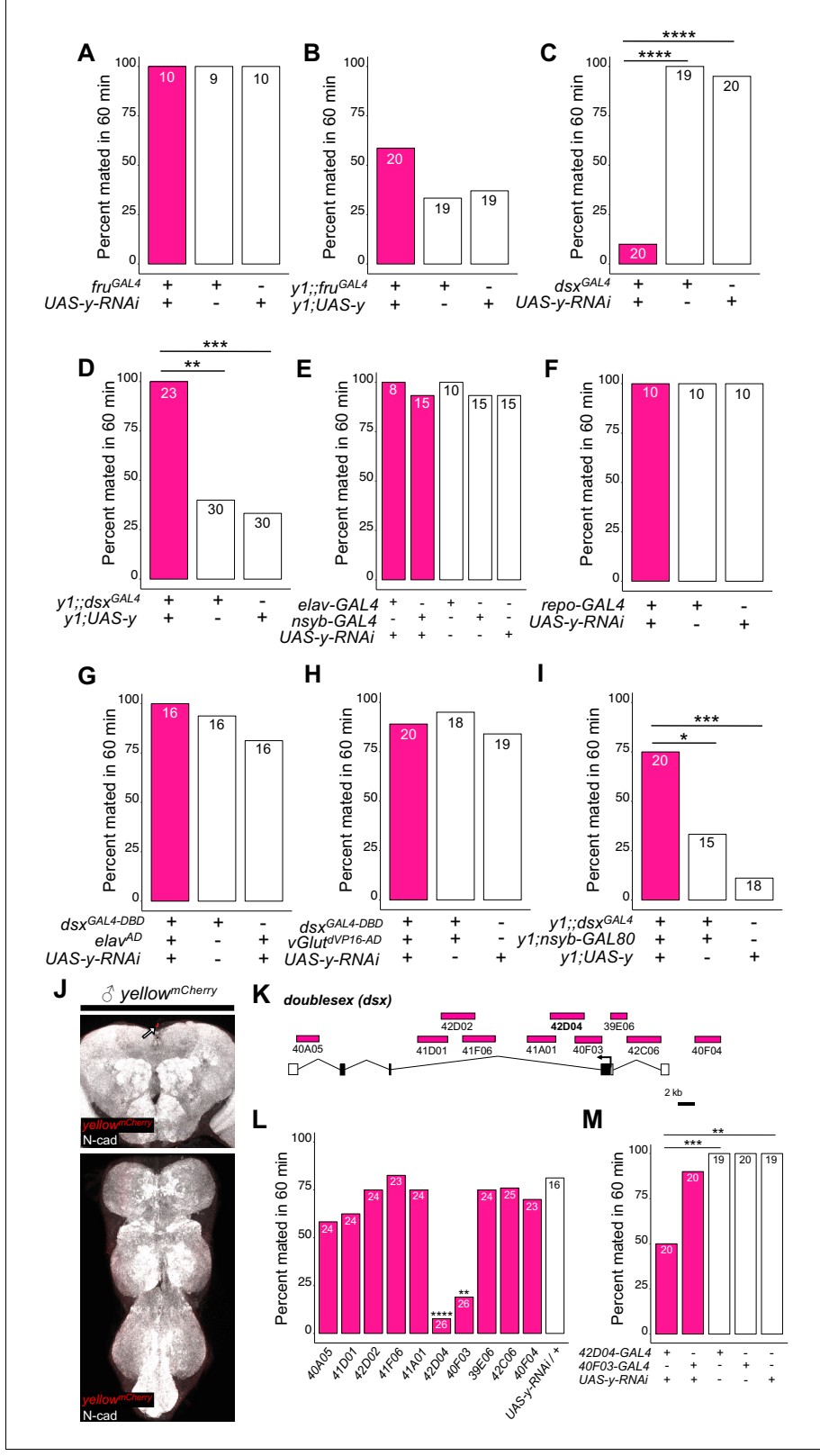

**Figure 2.** *yellow* expression in non-neuronal *doublesex*-expressing cells, but not *fruitless*-expressing cells, is necessary and sufficient for male mating success. (A,B) Neither expressing *yellow-RNAi* nor *yellow-cDNA* in *fru*-expressing cells using *fru^GAL4* (*Stockinger et al., 2005*) affected male copulation. (C) Expressing *yellow-RNAi* in *dsx*-expressing cells using *dsx^GAL4* (*Robinett et al., 2010*) significantly inhibited male mating success. (D)
*Figure 2 continued on next page*

*Figure 2 continued*

Expressing *yellow* in *dsx*-expressing cells using *dsx*$^{GAL4}$ in a *y*$^1$ mutant background was sufficient to restore male mating success. (E,F) Expressing *yellow-RNAi* using pan-neuronal (*elav-GAL4* and *nsyb-GAL4*) and pan-glia (*repo-GAL4*) drivers did not affect male mating success. (G) Restricting *yellow-RNAi* expression to *dsx*-expressing neurons using the split-GAL4 technique, combining *dsx*$^{GAL4-DBD}$ (*Pavlou et al., 2016*) with *elav*$^{VP16-AD}$ (*Luan et al., 2006*), did not affect male mating success. (H) Restricting *yellow-RNAi* expression to *dsx*-expressing glutamatergic neurons using the split-GAL4 technique, combining *dsx*$^{GAL4-DBD}$ (*Pavlou et al., 2016*) with *vGlut*$^{dVP16-AD}$ (*Gao et al., 2008*) did not affect male mating success. (I) Expressing *yellow* in *dsx*-expressing cells restricted outside the CNS using *dsx*$^{GAL4}$ and *nsyb-GAL80* (courtesy of Julie Simpson) in a *y*$^1$ mutant background significantly increased male mating success. (J) Brain and ventral nerve cord of adult male *y*$^{mCherry}$ flies stained with anti-N-Cadherin (N-cad) antibody labeling neuropil (white) and anti-DsRed antibody labeling Yellow::mCherry (red). We observed sparse, inconsistent signal outside the CNS at the top of the brain in males (white arrow), but we were unable to confirm a previous report that *y*$^{mCherry}$ is expressed in the adult brain (*Hinaux et al., 2018*). (K) Diagram of the male exon structure of the *dsx* locus highlighting 10 genomic fragments between 1.7 and 4 kb used to clone Janelia enhancer trap GAL4 drivers (*Pfeiffer et al., 2008*). Black boxes indicate coding exons. White boxes indicate 5' and 3' UTRs, and the arrow in exon two denotes the transcription start site. (L) Expressing *yellow-RNAi* using each Janelia *dsx-GAL4* driver identified *42D04-GAL4* and *40 F03-GAL4* as affecting male mating success when compared with the *yellow-RNAi* control. (M) A replicate experiment comparing *42D04-GAL4* and *40F03-GAL4* effects on male mating success with both GAL4 and UAS parental controls confirmed the significant effect of *42D04-GAL4* but not *40F03-GAL4*. We attribute differences in the *40F03-GAL4* effect between (L) and (M) to between experiment variability in the levels of male mating success; each common genotype tested in (L), for example, mated at higher levels in (M), but *42D04-GAL4* consistently showed a significant effect relative to controls. Sample sizes are shown at the top of each barplot. Significance was measured using Fisher's exact tests with Bonferroni corrections for multiple comparisons. Comparisons that were statistically (p<0.05) are indicated (\*p<0.05, \*\*p<0.01, \*\*\*p<0.001, \*\*\*\*p<0.0001).

DOI: https://doi.org/10.7554/eLife.49388.008

The following figure supplements are available for figure 2:

**Figure supplement 1.** *yellow* expression in *dsx*-expressing cells is necessary and sufficient for male mating success.

DOI: https://doi.org/10.7554/eLife.49388.009

**Figure supplement 2.** The mating regulatory sequence (MRS) from *Drapeau et al. (2006)* does not affect male mating success.

DOI: https://doi.org/10.7554/eLife.49388.010

**Figure supplement 3.** Expressing *yellow*-RNAi in subsets of CNS tissue does not affect male mating success.

DOI: https://doi.org/10.7554/eLife.49388.011

**Figure supplement 4.** *y*$^{mCherry}$ expression in adult female central nervous system.

DOI: https://doi.org/10.7554/eLife.49388.012

## *dsx*-expressing cells outside the CNS require yellow for normal male mating success

Although *dsx* is expressed broadly throughout the fly (*Robinett et al., 2010*; *Rideout et al., 2010*), we hypothesized that its expression in the nervous system would be responsible for *yellow*'s effects on mating because *yellow* has been reported to be expressed in the adult brain (*Hinaux et al., 2018*) and behavioral effects of other pigmentation genes are mediated by neurons (*Hotta and Benzer, 1969*; *Heisenberg, 1971*; *Borycz et al., 2002*; *True et al., 2005*). However, we found that suppressing *yellow* expression in the larval CNS, dopaminergic neurons, or serotonergic neurons (*Figure 2—figure supplement 3*, FET, P values ranging from 0.45 to 1), or in all neurons (*Figure 2E*, FET, p=1 in all cases) or all glia (*Figure 2F*, FET, p=1), had

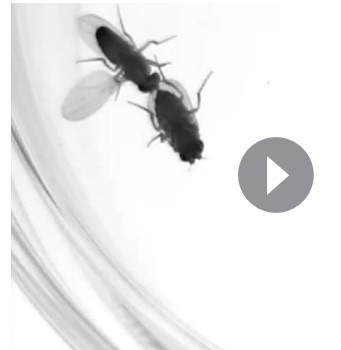

**Video 5.** Copulation attempts between male expressing *yellow-RNAi* in *dsx*$^{GAL4}$-expressing cells and wild-type female.

DOI: https://doi.org/10.7554/eLife.49388.013

no significant effect on male mating success. Specifically reducing *yellow* expression in either all *dsx*-expressing neurons (*Figure 2G*, FET, p=1 and 0.45) or all *dsx*-expressing glutamatergic neurons that are required for genital coupling (*Pavlou et al., 2016*) (*Figure 2H*, FET, p=1 and 0.68) also had no significant effect on male mating success. In addition, when we examined *yellow* expression in adult brains, we were only able to observe non-specific signal at the anterior of the adult brain in females (*Figure 2J*, *Figure 2—figure supplement 4*). Given this lack of evidence that *yellow* is required in neuronal cells for normal male mating behavior, we limited *dsx*$^{GAL4}$ activation of *yellow* expression in *y*$^1$ mutants to non-neuronal cells and found that these flies exhibited an increase in male mating success compared with *y*$^1$ mutant males (*Figure 2I*, FET, p=0.04 and 0.0002), showing that *yellow* expression in non-neuronal *dsx*-expressing cells is required for normal male mating behavior.

To identify which non-neuronal *dsx*-expressing cells require *yellow* expression for normal male mating success, we screened ten *dsx*-enhancer GAL4 lines that each contains a different ~ 3 kb region of *dsx* noncoding sequence (*Figure 2K*; *Pfeiffer et al., 2008*). Two of these lines, *42D04-GAL4* and *40F03-GAL4,* significantly decreased male mating success when driving *yellow-RNAi* (*Figure 2L*, FET, p=0.001 and 2 × 10$^{-5}$). These two GAL4 drivers contain overlapping sequences from intron 2 of *dsx* (*Figure 2K*), suggesting that their similar effects result from reduction of *yellow* expression in the same cells. Line *42D04-GAL4* had stronger effects than *40* F03-GAL4 (*Figure 2M*, FET, p=0.0009 for both controls for *42D04-GAL4* versus p=0.97 for both controls for *40* F03-GAL4), so we performed all further analyses with *42D04-GAL4*. Males with *yellow* reduced by *42D04-GAL4* performed courtship behavior in a pattern similar to *y*$^1$ mutant males: males performed all precopulatory courtship behaviors normally, but repeatedly failed to copulate, even after hours of attempts (*Video 6*). These data indicate that some or all cells in which *42D04-GAL4* drives expression require *yellow* expression for normal male mating behavior.

## Sex combs require *yellow expression for normal male mating success*

*42D04-GAL4* drives expression in a sexually dimorphic pattern in multiple neurons of the adult male (*Figure 3A,B*) and female CNS (*Figure 3—figure supplement 1A,B*), consistent with previously described *dsx*$^{GAL4}$ expression in the posterior cluster, the abdominal cluster, and, in males, in the prothoracic TN1 neurons (*Robinett et al., 2010*). *42D04-GAL4* also drives expression in male and female larval CNS and genital discs, with expression in the genital tissues persisting into the adult stage only in females (*Figure 3—figure supplement 1C–G*). Finally, we observed *42D04-GAL4* expression at the base of the sex combs (also observed by *Robinett et al., 2010* and *Rice et al., 2019*), which are modified bristles used during mating (*Cook, 1975*; *Ng and Kopp, 2008*; *Hurtado-Gonzales et al., 2015*) that are present only on the first tarsal segment of adult male forelegs (*Figure 3C–F*). Yellow protein is expressed in sex combs (*Hinaux et al., 2018*, *Figure 3G,H*), where it is presumably required for synthesis of black dopamine melanin in the sex comb 'teeth'. This expression of *yellow* in sex comb cells is driven by enhancer sequences in the *yellow* intron (*Figure 3—figure supplement 2*), potentially explaining why manipulating *yellow* expression using GAL4 driven by sequences 5' of the *yellow* gene failed to affect mating (*Figure 2—figure supplement 2A–D*). Driving expression of *yellow*-RNAi with *42D04-GAL4* eliminated expression of an mCherry tagged version of the native Yellow protein in sex combs and strongly reduced black melanin in the sex combs (*Figure 3I–L*) but not the abdomen (*Figure 3—figure supplement 1J*).

To test the impact of *yellow* expression in sex combs on male mating behavior, we used *42D04-GAL4* to drive *yellow-RNAi*, but inhibited the function of *42D04-GAL4* in the CNS with *nysb-GAL80* (courtesy of Julie Simpson). These flies showed no GAL4 activity in the CNS (*Figure 3M,N*), but lost black melanin in the sex combs (*Figure 3O*) and had reduced male mating success (*Figure 3P*, FET, p=0.002 and 0.08).

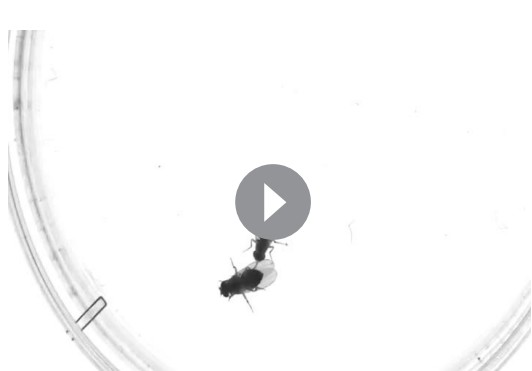

**Video 6.** Copulation attempts between male expressing *yellow-RNAi* in *42D04-GAL4*-expressing cells and wild-type female.
DOI: https://doi.org/10.7554/eLife.49388.014

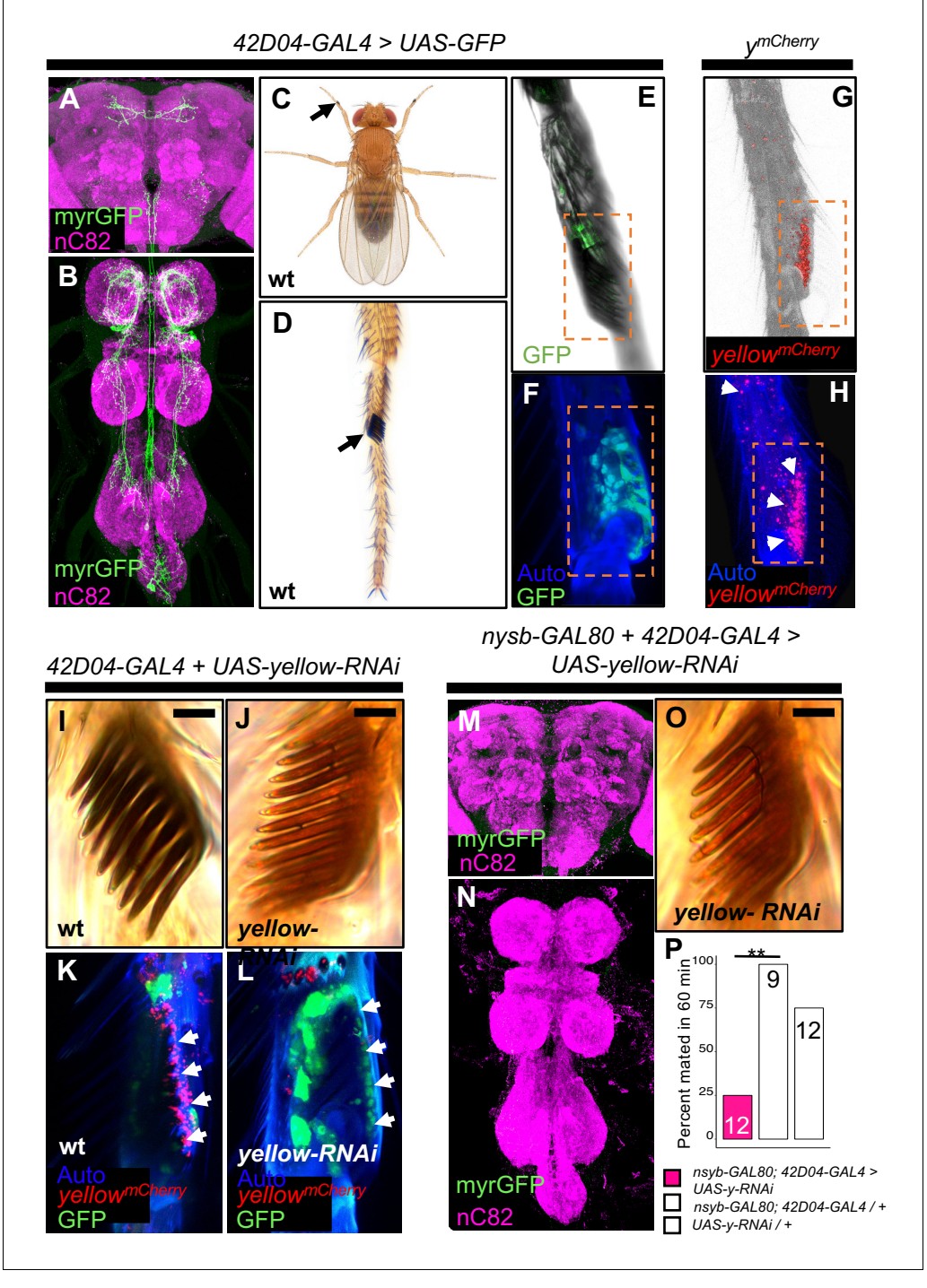

**Figure 3.** *yellow* expression in non-neuronal *42D04-GAL4* expressing cells is necessary for sex comb melanization and male mating success. (**A,B**) Brain and ventral nerve cord of adult male fly stained with anti-GFP (green) antibody for myrGFP expressed using *42D04-GAL4* and counterstained with anti-nC82 (magenta) for neuropil. (**C**) Wild-type (wt) *D. melanogaster* adult male fly highlighting the location of sex combs (Nicolas Gompel). (**D**) Close up of a wild-type (wt) sex comb on the first tarsal segment (ts1) of the front leg (courtesy of Nicolas Gompel). (**E**) Bright field illumination of a male front leg expressing cytGFP (green) in sex-comb cells using *42D04-GAL4*. (**F**) Confocal image of the sex comb cells expressing cytGFP (green) with *42D04-GAL4* and leg cuticle autofluorescence (blue). (**G**) Confocal image of a $y^{mCherry}$ male leg highlighting native $y^{mCherry}$ sex comb expression (red). (**H**) Zoomed in confocal image shown in (**G**) with leg cuticle autofluorescence (blue) and native $y^{mCherry}$ sex comb expression (red). (**I**) Wild-type (wt) sex comb. (**J**) Loss of black melanin in sex combs in males

*Figure 3 continued on next page*

*Figure 3 continued*

expressing *yellow-RNAi* using *42D04-GAL4*. (K) Co-localization of $y^{mCherry}$ (red) at the base of the sex comb cells expressing cytGFP (green) with *42D04-GAL4*. (L) Loss of $y^{mCherry}$ (red) at the base of the sex comb cells expressing cytGFP (green) and *yellow-RNAi* using *42D04-GAL4*. (M,N) Brain and ventral nerve cord of adult male expressing *nsyb-GAL80* to block GAL4 activity in the CNS, stained with anti-GFP (green) antibody for myrGFP expressed using *42D04-GAL4*, and counterstained with anti-nC82 (magenta) for neuropil. (O) Loss of black melanin in sex combs in *nsyb-GAL80* males expressing *yellow-RNAi* using *42D04-GAL4*. (P) Expressing *yellow-RNAi* using *42D04-GAL4* in males expressing *nsyb-GAL80* significantly inhibited male mating success. Scale bars in (I), (J), and (O) measure 12.5 μm. Sample sizes are shown at the top of each barplot. Significance was measured using Fisher's exact tests with Bonferroni corrections for multiple comparisons. Comparisons that were statistically (p<0.05) are indicated (**p<0.01).

DOI: https://doi.org/10.7554/eLife.49388.015

The following figure supplements are available for figure 3:

**Figure supplement 1.** Expression pattern of *42D04-GAL4*.
DOI: https://doi.org/10.7554/eLife.49388.016
**Figure supplement 2.** *yellow* EGFP reporters localize *yellow* sex comb expression to the intronic bristle enhancer.
DOI: https://doi.org/10.7554/eLife.49388.017

High-speed videos (1000 frames per second) revealed that *yellow* mutant ($y^1$) males fail repeatedly to grasp the female abdomen with their sex combs when attempting to mount and copulate (*Video 7*), whereas wild-type males more readily grasp the female with their melanized sex combs and initiate copulation efficiently (*Video 8*). These observations suggest that *yellow* expression in sex combs affects their melanization, which in turn affects their function.

## Sex comb melanization is required for efficient grasping, mounting and copulation

To test whether sex comb melanization (as opposed to some other unknown effect of losing *yellow* expression in sex combs) is critical for male sexual behavior, we suppressed expression of *Laccase2* (*Arakane et al., 2005*; *Riedel et al., 2011*) in sex combs using *42D04-GAL4* and *Laccase2-RNAi* (*Dietzl et al., 2007*). Laccase2 is required to oxidize dopamine into dopamine quinones and thus acts upstream of Yellow in the melanin synthesis pathway (*Figure 4A*; *Riedel et al., 2011*). Males with *Laccase2* suppressed in sex combs lacked both black and brown dopamine melanin, making these sex combs appear translucent (*Figure 4B*). These males displayed strongly reduced mating success compared with wild-type males (*Figure 4C*, FET, p=1×10$^{-7}$ and 8 × 10$^{-6}$) and behavioral defects similar to those observed for $y^1$ mutants (*Videos 9* and *10*), including inefficient grasping of the female for mounting and copulation. We noticed, however, that flies with *Laccase2-RNAi* driven by *42D04-GAL4* also showed a loss of melanin in the aedeagus (*Figure 4—figure supplement 1A*), which is the main part of the male genitalia used for copulation, despite no visible expression of *42D04-GAL4* in the adult male genitalia (*Figure 3—figure supplement 1G*) nor changes in aedeagus pigmentation in $y^1$ mutants (*Figure 4—figure supplement 1A*). We therefore used subsets of the *42D04* enhancer (*Figure 4—figure supplement 1B*) to drive expression of *Laccase2-RNAi*, separating the effects of expression in the sex combs from expression in the genitalia (*Figure 4—figure supplement 1C*). Male mating success was reduced when *Laccase2* suppression reduced melanization in the sex combs, but not the genitalia (*Figure 4—figure supplement 1D–G*).

How can sex comb melanization affect sex comb function? In insects, melanization impacts not only the color of the adult cuticle but also its mechanical stiffness (*Xu et al., 1997*;

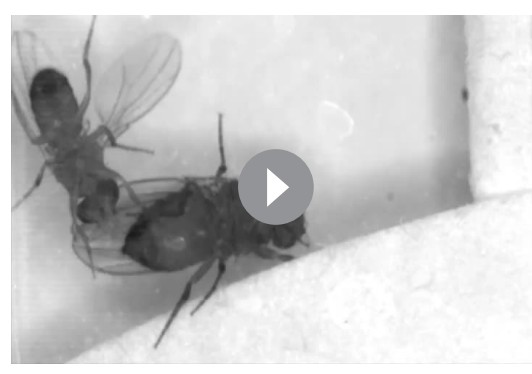

**Video 7.** High-speed (1000 fps) video capture of copulation attempts between $y^1$ male and wild-type female.
DOI: https://doi.org/10.7554/eLife.49388.018

*Kerwin et al., 1999*; *Vincent and Wegst, 2004*; *Andersen, 2005*; *Arakane et al., 2005*; *Suderman et al., 2006*; *Riedel et al., 2011*; *Noh et al., 2016*). For example, expressing *Laccase2-RNAi* in *D. melanogaster* wings softens the cuticle to such a degree that the wings collapse (*Riedel et al., 2011*). Butterflies lacking dopamine melanin due to loss of *yellow* or another gene required for melanin synthesis, *Dopa decarboxylase,* also show changes in the fine structure of their wing scales (*Matsuoka and Monteiro, 2018*). Consistent with these observations, we observed structural changes in *D. melanogaster* sex comb teeth lacking *yellow* or *Laccase2* expression using scanning electron microscopy (SEM), with a crack appearing in one of the *Laccase2-RNAi* comb teeth (*Figure 4D*). We thus conclude that these structural changes in sex combs are responsible for inhibiting the *yellow* mutant male's ability to grasp a female for mounting and copulation (*Video 10*). In 1976, *Wilson et al. (1976)* speculated about this very hypothesis based on their own observations of behavior in *yellow* mutant males.

Data from other *Drosophila* species are also consistent with this structural hypothesis. Specifically, *yellow* mutants in *D. subobscura*, *D. pseudoobscura*, and *D. gaucha*, all of which have sex combs, show reduced male mating success with wild-type females (*Rendel, 1944*; *Tan, 1946*; *Frias and Lamborot, 1970*; *Pruzan-Hotchkiss et al., 1992*) whereas *yellow* mutants in *Drosophila willistoni*, a species that lacks sex combs (*Kopp, 2011*; *Atallah et al., 2014*), do not (*da Silva et al., 2005*). Sex comb morphology is highly diverse among species that have sex combs (*Kopp, 2011*), but these structures generally seem to be melanized (*Figure 4—figure supplement 2*; *Tanaka et al., 2009*) and used to grasp females (*Videos 11–15*). (Our high-speed video recordings of mating in *D. anannasae*, *D. bipectinata*, *D. kikkawai*, *D. malerkotiana*, and *D. takahashi* show that differences in sex comb morphology (*Figure 4—figure supplement 2*) correspond with differences in how (where on the female and with which part of the male leg) the male grasps the female prior to copulation (*Videos 11–15*).

It remains unclear how *D. willistoni* males (and males of other species without sex combs) are able to efficiently grasp females prior to copulation (*Video 16*). However, differences in females might be part of the answer, as *D. melanogaster* $y^1$ mutant males are able to mate with $y^1$ mutant females at rates similar to wild-type males (*Bastock, 1956*; *Dow, 1976*; *Heisler, 1984*; *Liu et al., 2019*; *Figure 4—figure supplement 3A*, FET, p=1). That said, removing all melanin from *D. melanogaster* sex combs by knocking down *Laccase-2* reduced mating efficiency with $y^1$ females (*Figure 4—figure supplement 3B*, FET, p=0.02 and 0.0001), suggesting that the brown melanin remaining in $y^1$ sex-combs (*Figure 4B*) played a role in the mating success of $y^1$ males with $y^1$ females.

## Conclusion

Taken together, our data show that melanization of a secondary sexual structure affects mating in *D. melanogaster*. Specifically, we find that the reduced mating success of *D. melanogaster yellow* mutant males, which was perceived as a behavioral defect for decades, is caused by changes in the morphology of the structures used during mating. Other recent studies have also shown the importance of morphological structures for stickleback schooling (*Greenwood et al., 2015*), water strider walking (*Santos et al., 2017*), and cricket singing (*Pascoal et al., 2014*) behaviors. These observations all underscore that behavior cannot be understood by studying the nervous system alone; anatomy and behavior function and evolve as an interconnected system.

## Materials and methods

We have included a Key Resources Table as *Supplementary file 4*.

### Fly stocks and maintenance

The following lines were used for this work: $y^1$ [which was backcrossed into a wild-type (*Canton-S*) line for six generations before starting our experiments; the $y^1$ allele contains an A to C transversion in the ATG initiation and is considered a null allele (*Geyer et al., 1990*)]; *Canton-S* as wild-type (courtesy of Scott Pletcher); *UAS-yellow-RNAi* obtained from the Vienna Drosophila Resource Centre (VDRC) (*Dietzl et al., 2007*, KK106068); $y^1$;*UAS-y* (BDSC 3043); *elav-GAL4* (BDSC 49226); *nsyb-GAL4* (BDSC 39171); *repo-GAL4* (BDSC 7415); *dsx*$^{GAL4}$ (*Robinett et al., 2010*) (courtesy of Bruce Baker); *dsx*$^{GAL4}$ (*Rideout et al., 2010*) (courtesy of Stephen Goodwin); *fru*$^{GAL4}$ (*Stockinger et al., 2005*) (courtesy of Barry Dickson); the following Janelia enhancer trap GAL4 lines (*Pfeiffer et al.,*

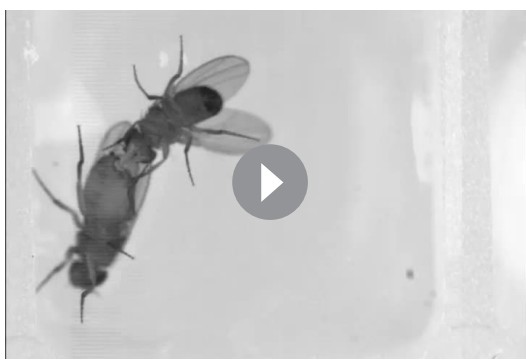

**Video 8.** High-speed (1000 fps) video capture of wild-type copulation.
DOI: https://doi.org/10.7554/eLife.49388.019

*2008*): *40A05-GAL4* (BDSC 48138), *41D01-GAL4* (BDSC 50123), *42D02-GAL4* (BDSC 41250), *41 F06-GAL4* (BDSC 47584), *41A01-GAL4* (BDSC 39425), *42D04-GAL4* (BDSC 47588), *40 F03-GAL4* (BDSC 47355), *39E06-GAL4* (BDSC 50051), *42 C06-GAL4* (BDSC 50150), *40 F04-GAL4* (BDSC 50094); $y^{mCherry}$ (courtesy of Nicolas Gompel); *nsyb-GAL80* (courtesy of Julie Simpson); *UAS-Laccase2-RNAi* obtained from the VDRC (*Dietzl et al., 2007*, KK101687); $dsx^{GAL4\text{-}DBD}$ (*Pavlou et al., 2016*) (courtesy of Stephen Goodwin); $vGlut^{dVP16\text{-}AD}$ (*Gao et al., 2008*) (courtesy of Stephen Goodwin); BDSC 6993; BDSC 49365; BDSC 6927; BDSC 45175; BDSC 3740; BDSC 5820; BDSC 8848; BDSC 7010; *TPH-GAL4* (courtesy of Shinya Yamamoto); *wing-body-GAL4* (BDSC 44373); *D. melanogaster yellow 5' up EGFP reporter* (*Kalay and Wittkopp, 2010*) (courtesy of Gizem Kalay); *D. melanogaster yellow intron EGFP reporter* (*Kalay and Wittkopp, 2010*) (courtesy of Gizem Kalay); *vasa-Cas9* (BDSC 51324); *UAS-cytGFP* (courtesy of Janelia Fly Core); *pJFRC12-10XUAS-IVS-myr::GFP* (courtesy of Janelia Fly Core). All flies were grown at 23°C with a 12 hr light-dark cycle with lights on at 8AM and off at 8PM on standard corn-meal fly medium.

## Behavior

### Mating assays

Virgin males and females were separated upon eclosion and aged for 3–8 d before each experiment. Experiments were carried out at 23°C on a 12 hr light dark cycle with lights on at 8 AM and off at 8 PM on standard corn-meal fly medium. Males were isolated in glass vials, and females were group housed in standard plastic fly vials at densities of 20–30 flies. All mating assays were performed at 23°C between 8-11AM or 6-9PM. For each assay replicate, a single virgin male and female fly were gently aspirated into a 35 mm diameter Petri dish (Genesee Scientific, catalog #32–103) placed on top of a 17 inch LED light pad (HUION L4S) and immediately monitored for 60 min for courtship and copulation activity. All genotypes tested initiated courtship (including tapping, chasing, wing extension, genital licking, and attempted copulation) towards the female. Any genotype that copulated within the 60 min window was scored as a successful mating. Except for the experiment described in *Figure 4—figure supplement 3* in which $y^1$ females were used, all females in mating assays were wild-type (*Canton-S*). The percent mated in 60 min values shown in figures were calculated as the number of replicates that mated divided by the total number of replicates and multiplied by 100.

### Courtship analysis

For courtship analysis, 60 min videos were recorded using Canon VIXIA HF R500 camcorders mounted to Manfrotto (MKCOMPACTACN-BK) aluminum tripods. To calculate courtship indices in *Figure 1* between wild-type and $y^1$ males, the amount of time males spent engaged in courtship: tapping, chasing, wing extension, genital licking, or attempted copulation was quantified for the first 10 min of the assay and divided by the total 10 min period. We chose to quantify courtship activity within the first 10 min of the assay, because wild-type (*Canton-S*) males will often begin copulating after this window, while $y^1$ males will continue to court throughout the entire 60 min period. Wing extension bouts were quantified by noting every unilateral wing extension bout for each genotype within the first 10 min of the assay.

### Song analysis

Courtship song was recorded as described previously (*Arthur et al., 2013*). All genotypes were recorded simultaneously. Song data were segmented (*Arthur et al., 2013*) and analyzed (http://

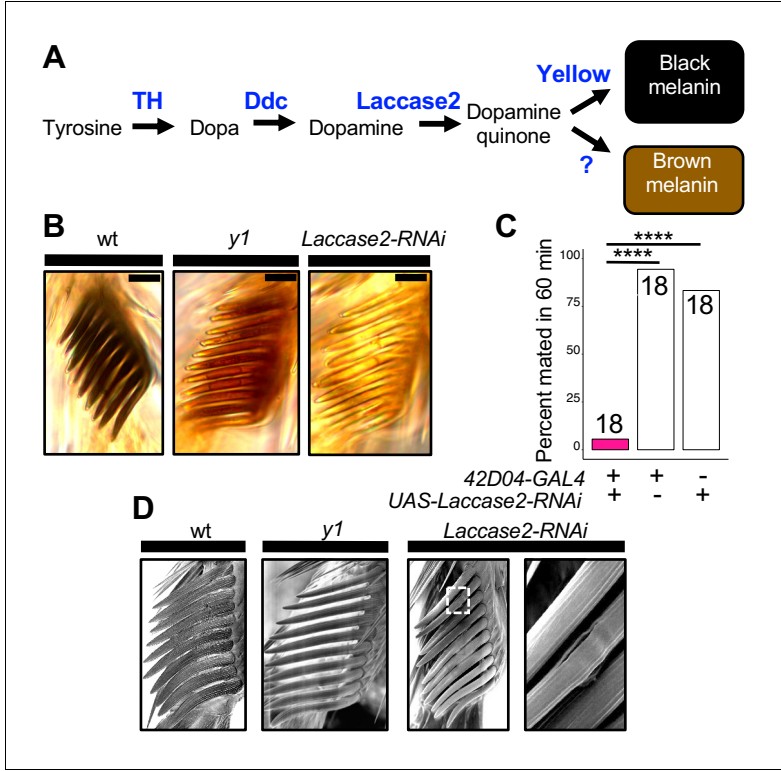

**Figure 4.** Sex comb melanization is specifically required for male mating success. (**A**) Simplified version of the insect melanin synthesis pathway. (**B**) Light microscopy images of sex combs from wild-type (wt), *y¹*, and *42D04-GAL4; UAS-Laccase2-RNAi* males. Expressing *Laccase2-RNAi* in sex combs completely blocked melanin synthesis. (**C**) Expressing *Laccase2-RNAi* using *42D04-GAL4* in males significantly inhibited male mating success. (**D**) Scanning Electron Microscopy (SEM) of sex combs from wild-type (wt), *y¹*, and *Laccase2-RNAi* males (expressed using *42D04-GAL4*). Compared to wild-type, sex comb teeth in *y¹* mutants appeared thinner and smoother, whereas *Laccase2-RNAi* sex comb teeth appeared even smoother than *y¹* mutants, and one comb tooth had a visible crack in the cuticle (white rectangle, enlarged on the right). Scale bars in (**B**) measure 12.5 μm. Sample sizes are shown at the top of each barplot. Significance in was measured using Fisher's exact tests with Bonferroni corrections for multiple comparisons. Comparisons that were statistically (p<0.05) are indicated (****p<0.0001).
DOI: https://doi.org/10.7554/eLife.49388.020

The following figure supplements are available for figure 4:

**Figure supplement 1.** Genetic dissection of the *42D04-GAL4* enhancer confirms the specific role of sex comb melanization, and not the aedeagus, in male mating success.
DOI: https://doi.org/10.7554/eLife.49388.021

**Figure supplement 2.** *Drosophila* species with varying sex comb morphology used for high-speed video assays.
DOI: https://doi.org/10.7554/eLife.49388.022

**Figure supplement 3.** Sex comb melanization is required for male mating success with *y¹* females.
DOI: https://doi.org/10.7554/eLife.49388.023

---

www.github.com/dstern/BatchSongAnalysis) without human intervention. Values for pulse per minute, sine per minute, and interpulse interval were then extracted from the software.

## High-speed video capture

For high-speed video capture of attempted mounting and copulation events, virgin males and females were isolated upon eclosion and aged for 4–7 d before each assay. Using a Fascam Photron SA4 (courtesy of Gwyneth Card) mounted with a 105 mm AF Micro Nikkor Nikon lens (courtesy of Gwyneth Card), we recorded individual pairs of males and females that were gently aspirated into a single well of a 96 well cell culture plate (Corning 05-539-200) partially filled with 2% agarose and covered with a glass coverslip. We recorded mounting and copulation attempts at 1000 frames per second (fps) and played back at 30 fps. Most wild-type males attempted mounting 3–5 times before

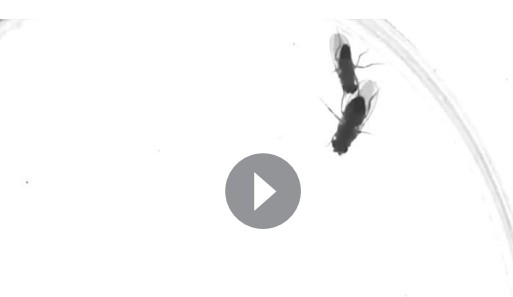

**Video 9.** Copulation attempts between male expressing *Laccase2-RNAi* in *42D04-GAL4*-expressing cells and wild-type female.
DOI: https://doi.org/10.7554/eLife.49388.024

copulating, whereas *y¹*, *yellow-RNAi*, and *Laccasse2-RNAi* males repeatedly attempted mounting without engaging in copulation, mirroring the videos we captured on the Canon VIXIA HF R500 at 30 fps.

## Imaging sex combs and genitalia
Sex comb images highlighting different melanization states (*Figure 3I,J,O*; *Figure 4B*) were taken using a Zeiss Axio Cam ERc 5 s mounted on a Zeiss Axio Observer A1 Inverted Microscope. Front legs were cut and placed sex comb side down on a microscope slide (Fisher brand 12-550-123) and imaged through a 40x objective. Images were processed using AxioVision LE software. Abdomens and genitalia images highlighting different melanization states of the aedeagus and female genital bristles were captured using a Canon EOS Rebel T6 camera mounted with a Canon MP-E 65 mm macro lens. Genitalia images were processed in Adobe Photoshop (version 19.1.5) (Adobe Systems Inc, San Jose, CA).

Focus Ion Beam Scanning Electron Microscope (FIB-SEM) images (*Figure 4D*) were taken by placing individual, dissected legs on carbon tape adhered to a scanning electron microscope pin stud mount with sex combs facing up. The samples were then coated with a 20 nm Au layer using a Gatan 682 Precision Etching and Coating System, and imaged by scanning electron microscopy in a Zeiss Sigma system. The samples were imaged using a 3-nA electron beam with 1.5 kV landing energy at 2.5MHz.

## Immunohistochemistry and confocal imaging
### Central Nervous System
Dissections, immunohistochemistry, and imaging of fly central nervous systems were done as previously described (*Aso et al., 2014*). In brief, brains and VNCs were dissected in Schneider's insect medium and fixed in 2% paraformaldehyde (diluted in the same medium) at room temperature for 55 min. Tissues were washed in PBT (0.5% Triton X-100 in phosphate buffered saline) and blocked using 5% normal goat serum before incubation with antibodies. Tissues expressing GFP were stained with rabbit anti-GFP (ThermoFisher Scientific A-11122, 1:1000) and mouse anti-BRP hybridoma supernatant (nc82, Developmental Studies Hybridoma Bank, Univ. Iowa, 1:30), followed by Alexa Fluor 488-conjugated goat anti-rabbit and Alexa Fluor 568-conjugated goat anti-mouse

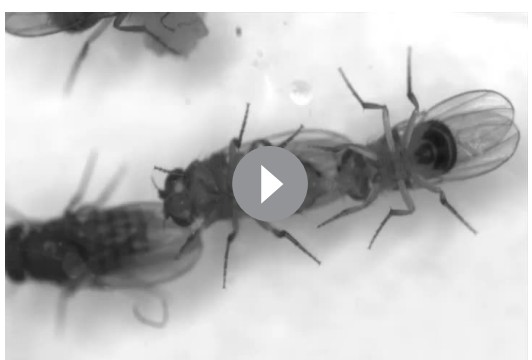

**Video 10.** High-speed (1000 fps) video capture of copulation attempts between male expressing *Laccase2-RNAi* in *42D04-GAL4*-expressing cells and wild-type female.
DOI: https://doi.org/10.7554/eLife.49388.025

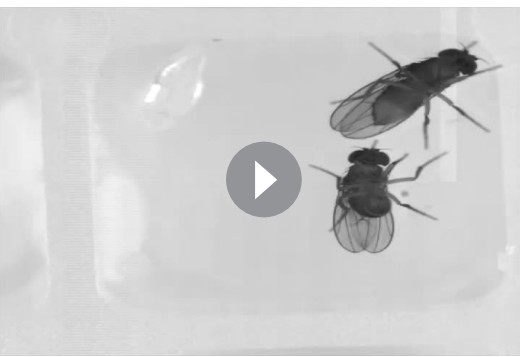

**Video 11.** *Drosophila anannasae* wild-type copulation.
DOI: https://doi.org/10.7554/eLife.49388.026

antibodies (ThermoFisher Scientific A-11034 and A-11031), respectively. Tissues expressing mCherry-tagged Yellow protein ($y^{mCherry}$) were stained with rabbit anti-dsRed (Clontech 632496, 1:1000) and rat anti-DN-Cadherin (DN-Ex #8, Developmental Studies Hybridoma Bank, Univ. Iowa, 1:100) as neuropil marker, followed by Cy3-conjugated goat anti-rabbit and Cy5-conjugated goat anti-rat antibodies (Jackson ImmunoResearch 111-165-144 and 112-175-167), respectively. After staining and post-fixation in 4% paraformaldehyde, tissues were mounted on poly-L-lysine-coated cover slips, cleared, and embedded in DPX as described. Image z-stacks were collected at 1 μm intervals using an LSM710 confocal microscope (Zeiss, Germany) fitted with a Plan-Apochromat 20x/0.8 M27 objective. Images were processed in Fiji (http://fiji.sc/) and Adobe Photoshop (version 19.1.5) (Adobe Systems Inc, San Jose, CA).

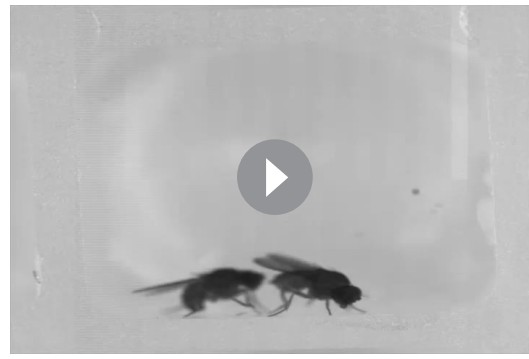

**Video 12.** *Drosophila bipectinata* wild-type copulation.
DOI: https://doi.org/10.7554/eLife.49388.027

## Sex combs and genitalia

Adult flies were 2–7 d old and pupae were 96 hr old after pupal formation (APF) for the EGFP reporter experiment summarized in *Figure 3—figure supplement 2*. Flies were anesthetized on ice, submerged in 70% ethanol, rinsed twice in phosphate buffered saline with 0.1% Triton X-100 (PBS-T), and fixed in 2% formaldehyde in PBS-T. Forelegs and genitalia/abdomen tips were removed with fine scissors and mounted in Tris-buffered (pH 8.0) 80% glycerol. Serial optical sections were obtained at 1.5 μm or 0.5 μm intervals on a Zeiss 880 confocal microscope with a LD-LCI 25x/0.8 NA objective (genitalia) or a Plan-Apochromat 40x/1.3 NA objective (appendages/tarsal sex combs). The native fluorescence of GFP, mCherry and autofluorescence of cuticle were imaged using 488, 594 and 633 lasers, respectively. Images were processed in Fiji (http://fiji.sc/), Icy (http://icy.bioimageanalysis.org/) and Adobe Photoshop (version 19.1.5) (Adobe Systems Inc, San Jose, CA).

## Generation of the mating regulatory sequence (MRS) deletion line

Using the 209 bp region mapped in *Drapeau et al. (2006)* between −300 and −91 bp upstream of *yellow*'s transcription start site, we designed two single guide RNA (gRNA) target sites at −291 bp and −140 bp that maximized the MRS deletion region, given constraints of identifying NGG PAM sites required for CRISPR/Cas9 gene editing (*Figure 2—figure supplement 2*). We in-vitro transcribed these gRNAs using a MEGAscript T7 Transcription Kit (Invitrogen) following the PCR-based protocol from *Bassett et al. (2013)*. Two 1 kb homology arms were PCR amplified from the *yellow* locus immediately upstream and downstream of the gRNA target sites using forward and reverse primers with NcoI and BglII tails, respectively, for the Left Arm (5'-TTACCATGGGGGATCAAG TTGAACCAC-3', 5'-GGAGATCTGGCCTTCA TCGACATTTA-3') and the forward and reverse primers with Bsu36I and MluI tails, respectively, for the Right Arm (5'-TACATCCCTAAGGCCTGA TTACCCGAACACT-3', 5'-TATACGCGTTGCCA TGCTATTGGCTTC-3') and cloned into pHD-DsRed-attp (*Gratz et al., 2014*; Addgene Plasmid # 51019) in two steps, digesting first with NcoI and BglII (Left Arm) to transform the Left Arm and second with Bsu36I and MluI (Right Arm) to transform the Right Arm, flanking the 3xP3::DsRed, attP, and LoxP sites. Homology

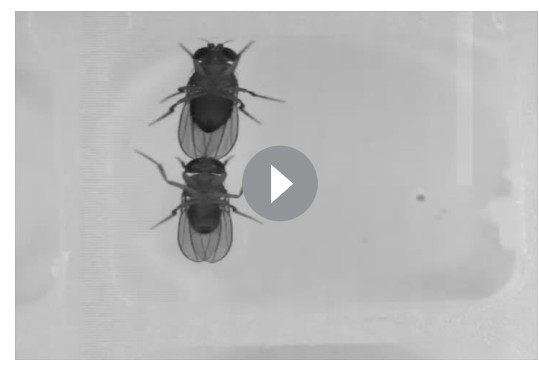

**Video 13.** *Drosophila kikkawai* wild-type copulation.
DOI: https://doi.org/10.7554/eLife.49388.028

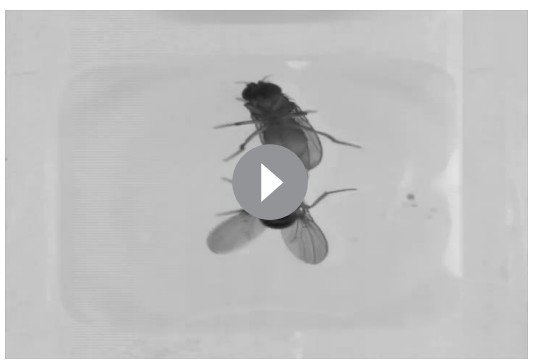

**Video 14.** *Drosophila malerkotiana* wild-type copulation.
DOI: https://doi.org/10.7554/eLife.49388.029

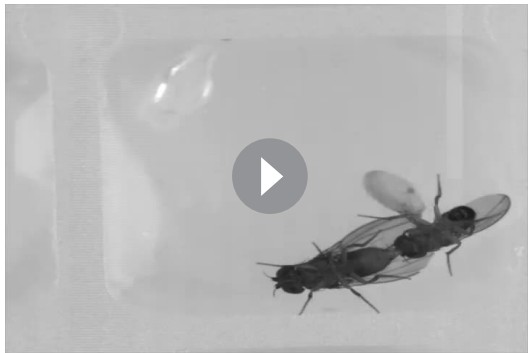

**Video 15.** *Drosophila takahashi* wild-type copulation.
DOI: https://doi.org/10.7554/eLife.49388.030

arms were ligated into pHD-DsRed-attp using T4 DNA Ligase (ThermoFisher Scientific), and products were transformed into One Shot TOP10 (Invitrogen) DH5 alpha competent cells. Purified donor plasmid was then co-injected at 500 ng/uL with the two gRNAs at 100 ng/uL total concentration into a *vasa-Cas9* (BDSC 51324) line. Flies were then screened for DsRed expression in the eyes, and Sanger sequenced verified for a 3xP3::DsRed replacement of the MRS region (***Figure 2—figure supplement 2***). We confirmed that we deleted 152 bp of the 209 bp region based on Sanger sequencing the CRISPR/Cas9 cut sites (***Figure 2—figure supplement 2***). Next, we crossed $y^{\Delta MRS+3xP3::DsRed}$ with a Cre-expressing fly line (courtesy of Bing Ye, University of Michigan) to excise 3xP3::DsRed and screened for flies that lost DsRed expression in the eyes. Finally, we PCR-gel verified that DsRed was indeed removed in creation of the $y^{\Delta MRS}$ line using the forward and reverse primers, respectively (5'-CAGTCGCCGATAAAGATGAACACTG-3', 5'-CAAGGTGATCAGGGTCACAAGGATC-3') (***Figure 2—figure supplement 2***).

## Generation of the 42D04-GAL4 enhancer sub-fragment pBPGUw lines

Enhancer sub-fragments (2 kb, 2 kb, 1.3 kb, 1.3 kb, and 1.3 kb for *42D04_A,B,C,D,E*-GAL4, respectively) were synthesized as IDT gene blocks (sequences available in ***Supplementary file 1***) based off of the 42D04 *D. melanogaster dsx* enhancer sequence (FBsf0000164494) (***Figure 4—figure supplement 1***). The gene blocks were designed with 5' and 3' Gibson tails to facilitate Gibson assembly (***Gibson et al., 2009***) into the GAL4 plasmid pBPGUw (***Pfeiffer et al., 2008***; Addgene Plasmid #17575) after digestion with FseI and AatII. Products were transformed into Mix and Go! DH5 alpha competent cells (Zymo). Clones were selected by ampicillin resistance on Amp-LB plates (60 mg/mL). Purified plasmids were injected at 500 ng/uL into the phiC31 integrase-expressing 86Fb landing site line *BDSC 24749* (courtesy of Rainbow Transgenics) for phiC31 attP-attB integration and screened for using a mini-white marker.

## Statistics

***Supplementary file 2*** is a Microsoft Excel file containing four worksheets with all of the data used for analysis. The worksheet labeled 'Univar_Male_Mating_Success_Data' contains a univariate description of each mating trial. The worksheet labeled 'Summary of mating success data' shows the number of successful and unsuccessful matings for each genotype tested (grouped by figure panel including the data) and was generated from the 'Univar_Male_Mating_-Success_Data' worksheet using the Excel Pivot Table function. The worksheet labeled 'Courtship_Data' includes the data for courtship index and wing extensions shown in ***Figure 1D and E***,

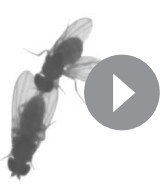

**Video 16.** *Drosophila willistoni* wild-type copulation.
DOI: https://doi.org/10.7554/eLife.49388.031

respectively. The worksheet labeled 'Song Data' includes the measures of pulses per minute, sin per minute and interpulse interval (labeled 'ModeEndToStartIPI') exported from the software described in *Arthur et al. (2013)*. R version 3.6.1 (2019-07-05) (*R Development Core Team, 2013*) was used for all statistical analyses using the code included in *Source code 1*. These analyses included *t*-tests comparing courtship index, number of wing extensions, pulses per minute, sine per minute, and interpulse interval that were run after exporting data in the Courtship Data and Song Data worksheets (separately) to tab delimited text files. Note that the default *t*-test parameters allowing for unequal variance between samples were used. *Source code 1* also contains the R code for the Fisher's Exact Tests, which were coded using data from the 'Summary of mating success data' worksheet. *Supplementary file 3* contains a summary of all statistical tests. Whenever an experimental genotype was compared to two control genotypes, P-values were adjusted using a Bonferroni correction for N = 2 (see *Supplementary file 3*). We note that for N = 2, alternative adjustments available with the p.adjust function in R ('holm', 'Hochberg', 'hommel' and 'fdr') give the same adjusted P-value.

## Acknowledgements

We thank members of the Wittkopp and Stern labs for helpful discussions. For fly strains, we thank Bruce Baker, Carmen Robinett, Stephen Goodwin, Barry Dickson, Scott Pletcher, Julie Simpson, Shinya Yamamoto, Bing Ye, Nicolas Gompel, Gizem Kalay, The Bloomington Drosophila Stock Center, The Vienna Drosophila RNAi Center, and the Janelia Fly Core. For fly injections, we thank Rainbow Transgenics Inc. For technical support with Scanning Electron Microscopy (SEM), we thank Harald Hess and Song Pang. For use of the Photron for high-speed video capture, we thank Gwyneth Card and W Ryan Williamson. CNS dissections, immunostaining, and imaging were performed by the Janelia Project Technical Resource team with special thanks to Gudrun Ihrke, Kari Close, and Christina Christoforou. We thank Nicolas Gompel, Abby Lamb, and Henry Ertl for comments on the manuscript. We also thank Shyama Nandakumar and Ajai Pulianmackal for help with dissections and confocal microscopy as well as Elena Kingston for capturing the *Drosophila willistoni* copulation video.

## Additional information

### Competing interests

Patricia J Wittkopp: Senior editor, *eLife*. The other authors declare that no competing interests exist.

### Funding

| Funder | Grant reference number | Author |
| --- | --- | --- |
| National Institutes of Health | T32GM007544 | Jonathan H Massey |
| National Institutes of Health | GM089736 | Patricia J Wittkopp |
| National Institutes of Health | 1R35GM118073 | Patricia J Wittkopp |

The funders had no role in study design, data collection and interpretation, or the decision to submit the work for publication.

### Author contributions

Jonathan H Massey, Conceptualization, Data curation, Formal analysis, Funding acquisition, Validation, Investigation, Visualization, Methodology, Writing—original draft, Writing—review and editing; Daayun Chung, Conceptualization, Data curation, Formal analysis, Validation, Investigation, Methodology, Writing—review and editing; Igor Siwanowicz, Resources, Data curation, Visualization, Methodology; David L Stern, Supervision, Funding acquisition, Visualization, Writing—original draft, Project administration, Writing—review and editing; Patricia J Wittkopp, Conceptualization, Supervision, Funding acquisition, Visualization, Writing—original draft, Project administration, Writing—review and editing

## Author ORCIDs

Jonathan H Massey ⓘD https://orcid.org/0000-0001-6182-2604
Igor Siwanowicz ⓘD http://orcid.org/0000-0001-5819-1530
David L Stern ⓘD https://orcid.org/0000-0002-1847-6483
Patricia J Wittkopp ⓘD https://orcid.org/0000-0001-7619-0048

## Decision letter and Author response

Decision letter https://doi.org/10.7554/eLife.49388.040
Author response https://doi.org/10.7554/eLife.49388.041

# Additional files

## Supplementary files

• Source code 1. R code used for statistical analyses.
DOI: https://doi.org/10.7554/eLife.49388.032

• Supplementary file 1. Sequences used for cloning.
DOI: https://doi.org/10.7554/eLife.49388.033

• Supplementary file 2. Summary table containing data for all mating trials.
DOI: https://doi.org/10.7554/eLife.49388.034

• Supplementary file 3. Summary of all statistical tests reported in the paper.
DOI: https://doi.org/10.7554/eLife.49388.035

• Supplementary file 4. Key Resources Table.
DOI: https://doi.org/10.7554/eLife.49388.036

• Transparent reporting form
DOI: https://doi.org/10.7554/eLife.49388.037

## Data availability

All data generated or analyzed during this study are included in the manuscript and supporting files. Supplementary File 2 contains all source data and Supplementary File 3 contains R code for analyzing it.

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
