## [Decision Letter]

Thank you for submitting your article "The yellow gene influences *Drosophila* male mating success through sex comb melanization" for consideration by *eLife*. Your article has been reviewed by two peer reviewers, and the evaluation has been overseen by K VijayRaghavan as the Senior and Reviewing Editor. The following individual involved in the review of your submission has agreed to reveal their identity: Catherine L Peichel (Reviewer #1).

The reviewers have discussed the reviews with one another and the Reviewing Editor has drafted this decision to help you prepare a revised submission.

Summary:

This is a beautiful study which identifies the genetic and mechanistic underpinnings of a component of male mating success in *Drosophila*. Through a rigorous and thorough analyses, this study overturns a long-held assumption that the yellow gene affects male courtship success due to its function in the nervous system. Rather, this study shows that yellow mutant males have lower mating success due to loss of pigment in the sex combs, which affects their structural integrity. The study is rigorously done, and the paper is clearly written and engaging. One reviewer said: I wish all the papers I reviewed were such a pleasure to read.

Essential revisions:

We have some concerns related to the presentation of the data. For example, song analysis is not mentioned in the Results/Discussion, but it is mentioned in the Materials and methods as well as being presented in the figures. Also, we do not think that the analysis of the MRS is as strong of evidence as it is presented, it is a negation of a past result rather than evidence in favor of a new conclusion. These could be attended to in the revision.

Secondly, we do not think it is absolutely necessary but the paper would have benefited from a fuller statistical model. For example, who was doing the observations of behaviour? Was it the same person every time? In general observer, date, time of day, etc. are important for behavioural comparisons and a full model including these variables is more appropriate than the simple tests described here. In addition, including actual p-values in the text would be welcome, as well as at least a mention of the test done, rather than having significance limited to stars in the figures. Showing the distribution of the data behind the bar charts would also be welcome. The major claims of the paper will be altered, but it would aide in transparent presentation of the data.

---

## [Author Response]

Essential revisions:We have some concerns related to the presentation of the data. For example, song analysis is not mentioned in the Results/Discussion, but it is mentioned in the Materials and methods as well as being presented in the figures. Also, we do not think that the analysis of the MRS is as strong of evidence as it is presented, it is a negation of a past result rather than evidence in favor of a new conclusion. These could be attended to in the revision.Secondly, we do not think it is absolutely necessary but the paper would have benefited from a fuller statistical model. For example, who was doing the observations of behaviour? Was it the same person every time? In general observer, date, time of day, etc. are important for behavioural comparisons and a full model including these variables is more appropriate than the simple tests described here. In addition, including actual p-values in the text would be welcome, as well as at least a mention of the test done, rather than having significance limited to stars in the figures. Showing the distribution of the data behind the bar charts would also be welcome. The major claims of the paper will be altered, but it would aide in transparent presentation of the data.

We very much appreciate the positive feedback and advice.

1) We have expanded the sentence to more precisely describe the data shown in the figure, as shown below:

“Fisher’s exact test, P = 6 x 10^-13^; Figure 1C). […] Watching the courtship videos showed that copulation initiation was most strikingly different between the two genotypes, with copulation initiation reduced in *yellow* males compared to wild-type (compare Videos 3 and 4).

2) We have softened the conclusions drawn from the MRS deletion as follows:

“We found that this deletion had no significant effect on male mating success (Figure 2—figure supplement 2H), contradicting the previous deletion mapping data (Drapeau et al., 2006). We conclude therefore that effects of *yellow* expression in *dsx*-expressing cells on mating behavior *are likely* mediated by other *cis*-regulatory sequences associated with the *yellow* gene.” Results from this experiment were not mentioned in the final conclusions section.

3) We have added a supplementary file (Supplementary file 2) to the manuscript that contains the raw data for every behavioral experiment in our study, including genotype, observer, date, time, and age information. Within each experiment, we performed assays on male/females that were aged consistently across every genotype. We also sampled at consistent time points across all genotypes and usually performed complete experiments on the same day. We performed logistic regression to test for effects of these variables, but did not include this analysis in the paper because for several of the experiments we could not compute contrasts for variables other than genotype because the ages and dates did not vary between genotypes.

The experiment with the most variability in date and age was the RNAi screen described in Figure 2M. To analyze data from this experiment, we imported the 260 observations into R and ran a logistic regression model with the glm (family = binary) function. Specifically, we tested for effects of genotype, date, and/or age on male mating success with the following model: glm(male_mating_success_60_min ~ genotype + date + male_age + female_age, family = "binomial"). A summary of the model fit is shown in Author response image 1. The only two significant predictors of male mating success in these data were genotype for the 40F03 GAL4 driver and the genotype for the 42D04 GAL4 driver. These are the same two (and only two) effects we detected as significant using the Fisher’s Exact tests described in Supplementary file 1.

4) We have incorporated exact P-values into the main text wherever it was possible to do so clearly. In addition, we’ve added Supplementary file 2 containing the data used for all statistical analyses, Supplementary file 3 containing the R code used to run the statistical tests, and Supplementary file 1, which summarizes all statistical tests. Note that in the prior version of the manuscript, Chi-squared tests were used to compare groups of genotypes followed by specific pairwise contrasts when statistically significant. In the revised manuscript, we have used Fisher’s exact tests to more directly compare each focal genotype to each control genotype within an experiment. For example, a genotype containing both UAS and GAL4 constructs is compared separately to controls containing only the UAS and only the GAL4 constructs. P-values from such a FET were then adjusted for the two tests by multiplying by 2, for a Bonferroni correction. We think that these FETs are more appropriate given our sample sizes and more directly test the contrasts between experimental and control genotypes. We note that this change in statistical testing did not change any of our conclusions.

5) The data for male mating success are binary (0 = mating failure, 1 = mating success), and the bar plots illustrate the proportions of successful matings in 60 minutes. There are thus no distributions of data to show. Wherever our data are continuous (e.g. Figure 1D-H), we have plotted the data rather than only present summary bar charts.